# The IL-10<sup>GFP</sup> (VeRT-X) mouse strain is not suitable for the detection of IL-10 production by granulocytes during lung inflammation

**Müge Özkan**[1], **Yusuf Cem Eskiocak**[2], **Gerhard Wingender**[2,3]*

1 Department of Genome Sciences and Molecular Biotechnology, Izmir International Biomedicine and Genome Institute, Dokuz Eylul University, Balcova/Izmir, Turkey, 2 Izmir Biomedicine and Genome Center (IBG), Balcova/Izmir, Turkey, 3 Department of Biomedicine and Health Technologies, Izmir International Biomedicine and Genome Institute, Dokuz Eylul University, Balcova/Izmir, Turkey

* gerhard.wingender@ibg.edu.tr

**Citation:** Özkan M, Eskiocak YC, Wingender G (2021) The IL-10<sup>GFP</sup> (VeRT-X) mouse strain is not suitable for the detection of IL-10 production by granulocytes during lung inflammation. PLoS ONE 16(5): e0247895. https://doi.org/10.1371/journal.pone.0247895

**Data Availability Statement:** All relevant data are within the manuscript and its Supporting Information files.

## Abstract

The clear and unequivocal identification of immune effector functions is essential to understand immune responses. The cytokine IL-10 is a critical immune regulator and was shown, for example, to limit pathology during various lung diseases. However, the clear identification of IL-10-producing cells is challenging and, therefore, reporter mouse lines were developed to facilitate their detection. Several such reporter lines utilize GFP, including the IL-10<sup>GFP</sup> (VeRT-X) reporter strain studied here. In line with previous reports, we found that this IL-10<sup>GFP</sup> line faithfully reports on the IL-10 production of lymphoid cells. However, we show that the IL-10<sup>GFP</sup> reporter is not suitable to analyse IL-10 production of myeloid cells during inflammation. During inflammation, the autofluorescence of myeloid cells increased to an extent that entirely masked the IL-10-specific GFP-signal. Our data illustrate a general and important technical caveat using GFP-reporter lines for the analysis of myeloid cells and suggest that previous reports on effector functions of myeloid cells using such GFP-based reporters might require re-evaluation.

## Introduction

Regulatory cytokines are important players during pulmonary inflammation to limit immunopathology without hampering effective pathogen clearance [1]. The regulatory cytokine IL-10 is involved in various lung diseases, like asthma, allergic airway disease (AAD), chronic obstructive pulmonary disease (COPD), and pulmonary infections [1]. IL-10 producing cells in the lung; FoxP3<sup>+</sup> and FoxP3<sup>-</sup> regulatory T cells [1], CD8<sup>+</sup> T cells [2], B cells, alveolar macrophages (AM) [3], interstitial macrophages (IM) [4], airway and alveolar epithelial cells [1], and airway-associated dendritic cells (DCs) [1] were shown to limit disease pathology. However, due to the low percentage of IL-10<sup>+</sup> cells within most cell populations and due to the low intensity of the flow cytometric staining of intracellular IL-10 the clear identification of IL-10 producing cells is challenging [5]. Therefore, various reporter mouse lines were developed to avoid the need for intracellular cytokine staining to identify IL-10 producing cells [6]. The IL-

**Funding:** This work was funded by grants from the Scientific and Technological Research Council of Turkey (TUBITAK, www.tubitak.gov.tr; #116Z272, GW), the Dokuz Eylul University (Izmir, Turkey; www.deu.edu.tr; #2020.KB.SAG.031, GW), and the European Molecular Biology Organization (EMBO, www.embo.org; IG3073; GW). The funders had no role in study design, data collection and analysis, decision to publish, or preparation of the manuscript.

**Competing interests:** The authors declare no competing financial interests.

**Abbreviations:** AAD, allergic airway disease; AM, alveolar macrophage; COPD, chronic obstructive pulmonary disease; DC, dendritic cell; HDM, house dust mite; ICCS, intracellular cytokine staining; ILC, innate lymphoid cell; IM, interstitial macrophages; GFP, green fluorescent protein; LPS, lipopolysaccharide; mLN, mediastinal lymph node.

10$^{GFP}$ (VeRT-X) strain studied here, expresses an (IRES)-enhanced green fluorescent protein (eGFP) fusion protein downstream of the exon 5 of the *il10* gene [7], and was reported to enable the identification of IL-10$^+$ lymphoid but also myeloid cells [7]. Several myeloid cells possess an autofluorescence around 525 nm [8], which coincides with the emission maxima of GFP around 530 nm [9]. However, the impact of this myeloid autofluorescence on the detection of GFP-reporter signals has not been fully clarified. Here, using an LPS-induced lung inflammation model, we demonstrate that the autofluorescence of myeloid cells conceals the IL10$^{GFP}$-specific signal. This was mainly due to a large increase in the myeloid autofluorescence during the inflammation. These data demonstrate that not all GFP-reporter mouse strains are suitable to analyse effector functions of myeloid cells during inflammation.

## Results and discussion

Neutrophils were suggested to be the primary source of IL-10 in the lung following *Klebsiella pneumoniae ST258* infection [10]. Furthermore, long-term exposure to house-dust mite (HDM) extracts suppressed allergic airway diseases via the IL-10 production by presumably FoxP3$^+$ T cells and alveolar macrophages [3]. In both studies, the 'shift' in the GFP-signal in the experimental IL-10$^{GFP}$ group was compared to either the uninfected IL-10$^{GFP}$ group or the naïve C57BL/6 group as control. When we analysed the IL-10 production during lung inflammation, we noticed that the background signal in the GFP-channel (505 nm—550 nm) of leukocytes from the lung, spleen, or mediastinal lymph node (mLN) was comparable in non-inflamed controls of either C57BL/6 (WT) or IL-10$^{GFP}$ mice (**Fig 1A**). Upon LPS-induced lung inflammation, the GFP-signal clearly increased, in particular in the lungs (**Fig 1B**). Nevertheless, the GFP-signal from WT and IL-10$^{GFP}$ mouse-derived cells remained comparable for the percentage (**Fig 1A and 1B**) or the mean fluorescence intensity (MFI, **Fig 1C**), irrespective of the organ analysed and the inflammatory condition. These data suggest that the GFP-signal detected in the IL-10$^{GFP}$—derived cells was actually a background signal.

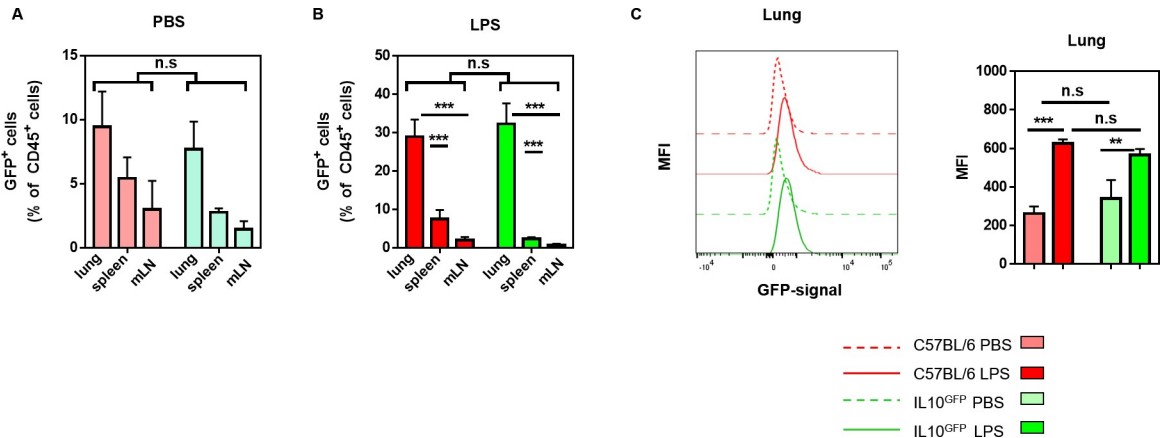

**Fig 1. The myeloid-derived GFP signal is comparable between IL10$^{GFP}$ and C57BL/6 mice and increases during lung inflammation.** C57BL/6 and IL-10$^{GFP}$ mice were challenged three times (d0, d1, d2) with either PBS or 10 μg LPS per mouse via aspiration. 16–18 hours after the last administration, the single cell suspension from lungs, spleens, and mLNs were stained for CD4$^+$ T cells (CD45$^+$ CD45R/B220$^-$ CD3ε$^+$ CD4$^+$ CD8α$^-$), CD8$^+$ T cells (CD45$^+$ CD45R/B220$^-$ CD3ε$^+$ CD4$^-$ CD8α$^+$), macrophages (CD45$^+$ CD45R/B220$^-$ CD3ε$^-$ Siglec-F$^-$ F4/80$^+$), Siglec-F$^+$ cells (eosinophils, alveolar macrophages; CD45$^+$ CD45R/B220$^-$ CD3ε$^-$ Siglec-F$^+$ F4/80$^{-/+}$), neutrophils (CD45$^+$ CD45R/B220$^-$ CD3ε$^-$ Siglec-F$^-$ F4/80$^-$ Ly6G$^+$ CD11b$^+$), monocytes (CD45$^+$ CD45R/B220$^-$ CD3ε$^-$ Siglec-F$^-$ F4/80$^-$ Ly6G$^-$ Ly6C$^+$ CD11b$^+$), and ILCs (CD45$^+$ CD45R/B220$^-$ CD3ε$^-$ Siglec-F$^-$ F4/80$^-$ Ly6G$^-$ CD90.2$^+$ CD127$^{lo/-}$). The expression of IL-10 was measured either by GFP$^+$ or by intracellular IL-10 as indicated. **(A, B)** The frequency of GFP$^+$ cells (live CD45$^+$) in the lungs of (A) PBS or (B) LPS challenged C57BL/6 (red) and IL-10$^{GFP}$ (green) mice are shown. The graph shows combined data from three independent experiments (PBS: n = 9 mice/group, LPS: n = 13–15 mice/group in total). **(C)** Representative histogram showing the shift in the GFP-signal (505 nm —550 nm) of live CD45$^+$ lung cells upon LPS administration in C57BL/6 and IL-10$^{GFP}$ mice strains.

To clarify which cell types could produce IL-10, we directly compared the IL-10 signal derived from the GFP-signal or from intracellular cytokine staining (ICCS) for CD4$^+$ and CD8$^+$ T cells, neutrophils, macrophages, Siglec-F$^+$ cells (eosinophils, alveolar macrophages), monocytes, and ILCs. Cells from the lungs, spleens, and mLN of PBS (control) and LPS-challenged C57BL/6 and IL-10$^{GFP}$ mice were analysed (S1 Fig). Although intracellular IL-10 staining is challenging, we obtained a clear IL-10 signal with the commercial antibody (S2 Fig). With this side-by-side comparison, we found that the IL-10-signal derived from GFP or ICCS correlated well for CD4$^+$ (Fig 2A) and CD8$^+$ T cells (Fig 2B). These data indicate that the IL-10GFP—signal faithfully reports on the IL-10 production of lymphoid cells. However, the GFP-signal from neutrophils (Fig 2C), macrophages (Fig 2D), and Siglec-F$^+$ (Fig 2E) cells was significantly higher than the ICCS-derived signal. Representative dot-plots for all cell types and organs are provided in S3 Fig. Furthermore, the additional staining with a secondary αGFP-AF488-antibody was not able to improve the specificity of the IL-10-signal (S4 Fig). Interestingly, the GFP-signal of monocytes (Fig 2F) and ILCs (Fig 2G) was substantially higher only in the lung tissues, indicating that the increase in the GFP-channel autofluorescence is organ-specific for some cell types.

An overlay of the GFP-signals of CD4$^+$ T cells, CD8$^+$ T cells, neutrophils, macrophages, Siglec-F$^+$ cells, monocytes, and ILCs from control C57BL/6 lung (Fig 3A) and spleen (Fig 3B) indicated that the background GFP-signal was mainly derived from granulocytes. According to the background GFP-signal, cell types ranked as Siglec-F > macrophage > neutrophil > monocyte > ILC. A similar increase in the GFP-channel autofluorescence upon inflammation was noted when alveolar macrophages were analysed (Fig 4).

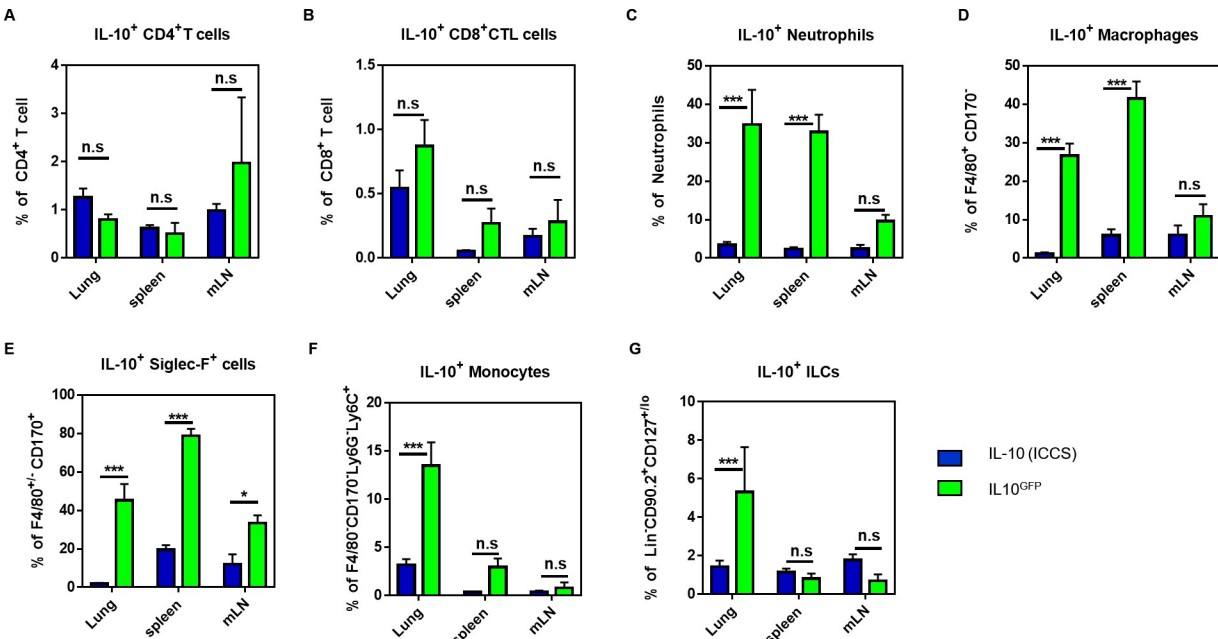

**Fig 2. The myeloid-derived IL-10$^{GFP}$ signal is masked by autofluorescence in granulocytes.** C57BL/6 and IL-10$^{GFP}$ mice were challenged three times (d0, d1, d2) with either PBS or 10 μg LPS per mouse via aspiration. 16–18 hours after the last administration, the single cell suspension from lungs, spleens, and mLNs were stained for CD4$^+$ T cells (CD45$^+$ CD45R/B220$^-$ CD3ε$^+$ CD4$^+$ CD8α$^-$), CD8$^+$ T cells (CD45$^+$ CD45R/B220$^-$ CD3ε$^-$ CD4$^-$ CD8α$^+$), macrophages (CD45$^+$ CD45R/B220$^-$ CD3ε$^-$ Siglec-F$^-$ F4/80$^+$), Siglec-F$^+$ cells (eosinophils, alveolar macrophages; CD45$^+$ CD45R/B220$^-$ CD3ε$^-$ Siglec-F$^+$ F4/80$^{-/+}$), neutrophils (CD45$^+$ CD45R/B220$^-$ CD3ε$^-$ Siglec-F$^-$ F4/80$^-$ Ly6G$^+$ CD11b$^+$), monocytes (CD45$^+$ CD45R/B220$^-$ CD3ε$^-$ Siglec-F$^-$ F4/80$^-$ Ly6G$^-$ Ly6C$^+$ CD11b$^+$), and ILCs (CD45$^+$ CD45R/B220$^-$ CD3ε$^-$ Siglec-F$^-$ F4/80$^-$ Ly6G$^-$ CD90.2$^+$ CD127$^{lo/-}$). The expression of IL-10 was measured either by GFP$^+$ or by intracellular IL-10 as indicated. **(A-G)** Comparison of the IL-10 signal derived from intracellular cytokine staining (ICCS) or from the GFP-signal (IL10$^{GFP}$) of IL10$^{GFP}$ mice in (D) CD4$^+$ T cells, (E) CD8$^+$ T cells, (F) neutrophils, (G) macrophages, (H) Siglec-F$^+$ cells (eosinophils, alveolar macrophages), (I) monocytes, and (J) ILCs from indicated organs. The graph shows combined data from two independent experiments (PBS: n = 6 mice/group, LPS: n = 7–9 mice/group in total).

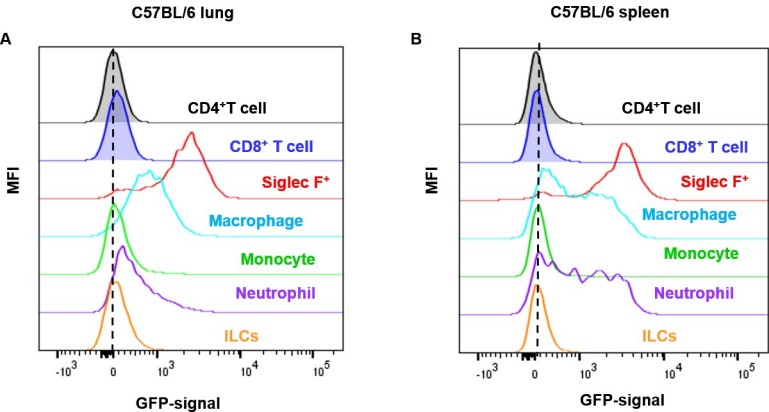

**Fig 3. The cell-type-specific autofluorescence in C57BL/6 mice in control conditions.** Representative histograms from the **(A)** lung and the **(B)** spleen of PBS challenged C57BL/6 mice, demonstrating the increase in autofluorescence in the GFP-channel for myeloid cells. The graphs show representative data from three independent experiments (PBS: n = 9 mice/group, LPS: n = 13–15 mice/group in total).

In conclusion, our data indicate that the IL-10$^{GFP}$ (VeRT-X) reporter strain is not suitable to analyse IL-10 production of myeloid cells during inflammation, due to the strong increase in the autofluorescence, which masks the IL-10-specific GFP-signal. Although we only analysed lung, spleen, and mLN of the VeRT-X IL-10$^{GFP}$ reporter strain, it appears likely that this problem will also be relevant to other organs. Importantly, the strong increase of the granulocytic autofluorescence in the GFP-channels was independent of the strain analysed. Therefore, it is expected that other fluorescent reporter lines that utilize the GFP-channel would face similar problems at distinguishing the reporter-specific signal from the autofluorescence signal when analysing granulocytes. Our data indicate that granulocytes are the source of the background GFP-signal, in line with previous publications reporting high levels of autofluorescence [11], although other sources, like collagen deposition [12,13], cannot be excluded. This large increase in false-positive GFP-signals for myeloid cells during inflammation could have been missed previously due to a focus on lymphoid cells or due to a lack of the WT-treated controls. Our data illustrate a general and important technical caveat using GFP-reporter lines

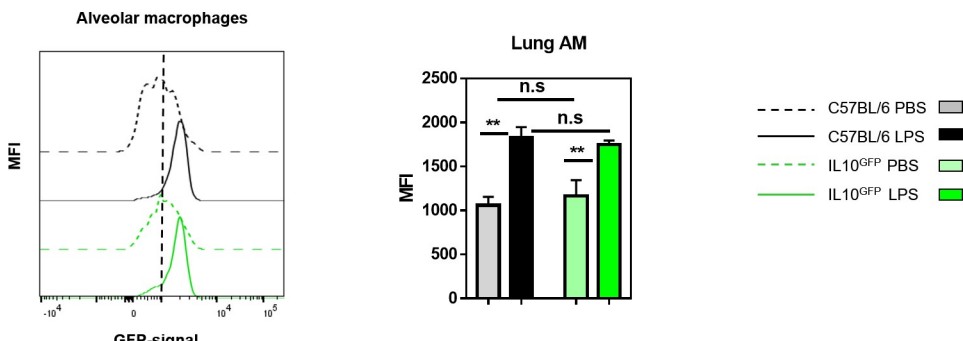

**Fig 4. The autofluorescence of alveolar macrophages increases during lung inflammation.** C57BL/6 and IL-10$^{GFP}$ mice were subjected to three daily administrations (d0, d1, d2) of 10 µg LPS per mouse via aspiration. 16–18 hours after the last administration, the single-cell suspensions from the lungs were analysed. Representative histogram showing the GFP-signal from alveolar macrophages (CD45$^+$ CD45R/B220$^-$ CD3ε$^-$ Siglec-F$^+$ F4/80$^+$ CD11b$^-$) from C57BL/6 and IL10$^{GFP}$ mice as indicated. The data shown are representative of three independent experiments (PBS: n = 9 mice/group, LPS: n = 13–15 mice/group in total).

for the analysis of myeloid cells and suggest that previous reports on effector functions of myeloid cells using such GFP-based reporters might require re-evaluation.

## Material and methods

### Mice

C57BL/6J mice and IL-10$^{\text{GFP}}$ (also known as Vert-X) B6(Cg)-Il10$^{\text{tm1.1Karp}}$/J mice were originally obtained from Jackson Laboratory (Bar Harbor, USA). All mice were housed in the vivarium of the Izmir Biomedicine and Genome Center (IBG, Izmir, Turkey) in accordance with the respective institutional animal care committee guidelines. All mouse experiments were performed with prior approval by the institutional ethic committee ('Izmir Biomedicine and Genome Center's Ethical Committee on Animal Experimentation'), in accordance with national laws and policies. All the methods were carried out in accordance with the approved guidelines and regulations.

### Reagents and monoclonal antibodies

Monoclonal antibodies against the following mouse antigens were used in this study: CD3ε (145.2C11), CD4 (RM4-5), CD8α (53–6.7), CD11b (M1/70), CD11c (N418), CD45 (30-F11), CD45R/B220 (RA3-6B2), CD90.2 (30-H12), CD127 (A7R34), CD170/Siglec-F (E50-2440), F4/80 (BM8), IL-10 (JESS-16E3), Ly6C (HK1.4), Ly6G (1A8). Antibodies were purchased from BD Biosciences (San Diego, USA), BioLegend (San Diego, USA), or ThermoFisher Scientific (Carlsbad, USA). Antibodies were conjugated to Pacific Blue, Brilliant Violet 421, V500, Brilliant Violet 510, Brilliant Violet 570, Brilliant Violet 650, Brilliant Violet 711, Brilliant Violet 785, Brilliant Violet 786, FITC, Alexa Fluor 488, PerCP-Cy5.5, PerCP-eFluor 710, PE, PE-CF594, PE-Cy7, APC, Alexa Fluor 647, Alexa Fluor 700, APC-Cy7, or APC-Fire750. Details on the antibody used in this study are given in the **S1 Table**. Anti-mouse CD16/32 antibody (2.4G2) and Zombie UV Dead Cell Staining Kit were obtained from Tonbo Biosciences (San Diego, USA) and from BioLegend, respectively. Unconjugated mouse and rat IgG antibodies were purchased from Jackson ImmunoResearch (West Grove, USA).

### LPS-induced lung inflammation

Lung inflammation was induced by three daily administrations (d0, d1, d2) with 10 μg LPS (#L6386, Sigma-Aldrich, St. Louis, USA) per mouse via pharyngeal/laryngeal aspiration. 16–18 hours after the last challenge, the mice were sacrificed, and lungs, spleens, and mediastinal lymph nodes were collected.

### Cell preparation

Lungs were removed and minced into smaller pieces in a 6-well plate (Greiner, Germany). The digestion mixture, composed of 1 mg/mL collagenase D and 0.1 mg/mL DNase I (both from Roche, Switzerland) in complete RPMI medium (Gibco, USA), was added to the samples and incubated for 45 min at 37°C on a lateral shaker. The lung samples were filtered through 100 μm mesh with PBS, washed twice, and the red blood cells were eliminated by ACK lysis buffer (Lonza, USA). Spleens and mediastinal lymph nodes were homogenized by filtering through a 76 μm mesh with ice-cold PBS (Lonza), washed twice, and red blood cells were eliminated by ACK lysis buffer (Lonza).

## *In vitro* stimulation

Lymphocytes were stimulated *in vitro* with PMA (50 ng/mL) and ionomycin (1 μg/mL) (both Sigma-Aldrich, St. Louis, MO) for four hours at 37˚C in the presence of both Brefeldin A (GolgiPlug) and Monensin (GolgiStop) in complete RPMI medium (RPMI 1640 medium (Life Technologies); supplemented with 10% (v/v) fetal calf serum (FCS), 1% (v/v) Pen-Strep-Glutamine (10.000 U/ml penicillin, 10.000 μg/ml streptomycin, 29.2 mg/ml L-glutamine (Life Technologies)) and 50 μM β-mercaptoethanol (Sigma)). As GolgiPlug and GolgiStop (both BD Biosciences) were used together, half the amount recommended by the manufacturer were used.

## Flow cytometry

Flow cytometry was performed as described [14]. In brief, for staining of cell surface molecules, cells were suspended in staining buffer (PBS, 1% BSA, 0.01% NaN$_3$) and stained with fluorochrome-conjugated antibodies (0.1–1 μg/10$^6$ cells, or according to the manufacturer's recommendations) for 15 min in a total volume of 50 μl at 4˚C for 30 min. FcεR-blocking antibody αCD16/32 (2.4G2) and unconjugated rat and mouse IgG (Jackson ImmunoResearch) were added to prevent non-specific binding. If biotin-conjugated antibodies were used, cell-bound antibodies were detected with streptavidin conjugates (1:200, or according to the manufacturer's recommendations) in a second incubation step. Dead cells were labelled with a commercially available Zombie UV Dead Cell Staining Kit (BioLegend). For the analysis of intracellular cytokines, cells were fixed and permeabilized with Cytofix/Cytoperm (BD Biosciences) for 10 min at 37˚C. Cells were washed twice and incubated overnight at 4˚C with the fluorochrome-conjugated antibodies and unconjugated rat and mouse IgG in Perm/Wash solution (BD Biosciences), which was followed by additional 5 min incubation in Perm/Wash solution without antibodies. For the analysis of the GFP signal, the cells were fixed with freshly prepared 2% formaldehyde solution for 40 minutes on ice as described [15]. This fixation allows the detection of the GFP-signal together with other intracellular proteins [15]. Then, the cells were permeabilized using the BD Perm/Wash solution. Cells were washed twice and incubated overnight at 4˚C with the fluorochrome-conjugated antibodies and unconjugated rat and mouse IgG in Perm/Wash solution, which was followed by additional 5 min incubation in Perm/Wash solution without antibodies. Cells were analysed with FACSCanto or LSR-Fortessa (BD Biosciences), and data were processed with CellQuest Pro (BD Biosciences) or Flow Jo (Tree Star) software.

## Statistical analysis

Data are presented as mean ± standard error of the mean (SEM). The statistical analysis was performed with GraphPad Prism 7.0 software (GraphPad Software, San Diego, USA). One-way ANOVA followed by Holm-Sidak posthoc test are used to compare p-values regarded as *p≤0.05, **p≤0.01, and ***p≤0.001.

## Supporting information

**S1 Fig. Gating strategy to identify leukocytes from the lung, spleen, and mLN.** A graphic outline (**A**) and exemplary graphs (**B**) are given to illustrate the gating strategy employed to identify CD4$^+$ T cells (CD45$^+$ CD45R/B220$^-$ CD3ε$^+$ CD4$^+$ CD8α$^-$ CD127$^{lo}$), CD8$^+$ T cells (CD45$^+$ CD45R/B220$^-$ CD3ε$^+$ CD4$^-$ CD8α$^+$), macrophages (CD45$^+$ CD45R/B220$^-$ CD3ε$^-$ Siglec-F$^-$ F4/80$^+$), Siglec-F$^+$ cells (eosinophils, alveolar macrophages; CD45$^+$ CD45R/B220$^-$ CD3ε$^-$ Siglec-F$^+$ F4/80$^{-/+}$), neutrophils (CD45$^+$ CD45R/B220$^-$ CD3ε$^-$ Siglec-F$^-$ F4/80$^-$ Ly6G$^+$

CD11b$^+$), monocytes (CD45$^+$ CD45R/B220$^-$ CD3ε$^-$ Siglec-F$^-$ F4/80$^-$ Ly6G$^-$ Ly6C$^+$ CD11b$^+$), and ILCs in the inflamed lung.
(TIF)

**S2 Fig. Efficiency of intracellular IL-10 cytokine staining.** Representative flow cytometric dot plots showing the IL-10 staining (ICCS) in splenic CD4$^+$ T cells.
(TIF)

**S3 Fig. Representative flow cytometry plots for the data presented in Fig 2.** C57BL/6 and IL-10$^{GFP}$ mice were challenged three times (d0, d1, d2) with either PBS or 10 μg LPS per mouse via aspiration. 16–18 hours after the last administration, the single cell suspension from **(A)** lungs, **(B)** spleens, and **(C)** mLNs were stained for CD4$^+$ T cells (CD45$^+$ CD45R/B220$^-$ CD3ε$^+$ CD4$^+$ CD8α$^-$), CD8$^+$ T cells (CD45$^+$ CD45R/B220$^-$ CD3ε$^+$ CD4$^-$ CD8α$^+$), macrophages (CD45$^+$ CD45R/B220$^-$ CD3ε$^-$ Siglec-F$^-$ F4/80$^+$), Siglec-F$^+$ cells (eosinophils, alveolar macrophages; CD45$^+$ CD45R/B220$^-$ CD3ε$^-$ Siglec-F$^+$ F4/80$^{-/+}$), neutrophils (CD45$^+$ CD45R/B220$^-$ CD3ε$^-$ Siglec-F$^-$ F4/80$^-$ Ly6G$^+$ CD11b$^+$), monocytes (CD45$^+$ CD45R/B220$^-$ CD3ε$^-$ Siglec-F$^-$ F4/80$^-$ Ly6G$^-$ Ly6C$^+$ CD11b$^+$), and ILCs (CD45$^+$ CD45R/B220$^-$ CD3ε$^-$ Siglec-F$^-$ F4/80$^-$ Ly6G$^-$ CD90.2$^+$ CD127$^{lo/-}$). The expression of IL-10 was measured either by intracellular IL-10 staining (left panel) or GFP—expression (right panel).
(TIF)

**S4 Fig. The staining with a secondary αGFP-AF488 antibody does not improve the resolution of the myeloid IL10$^{GFP}$ signal.** Representative histograms from the spleen of PBS (left) or LPS (right) challenged C57BL/6 and IL-10$^{GFP}$ mouse, demonstrating the overall GFP-signal detected with or without labelling with secondary αGFP-AF488.
(TIF)

**S1 Table.**
(TIF)

## Author Contributions

**Conceptualization:** Müge Özkan, Gerhard Wingender.

**Data curation:** Müge Özkan.

**Formal analysis:** Müge Özkan, Yusuf Cem Eskiocak.

**Funding acquisition:** Gerhard Wingender.

**Investigation:** Müge Özkan, Yusuf Cem Eskiocak, Gerhard Wingender.

**Methodology:** Müge Özkan, Yusuf Cem Eskiocak, Gerhard Wingender.

**Project administration:** Müge Özkan, Gerhard Wingender.

**Resources:** Gerhard Wingender.

**Supervision:** Gerhard Wingender.

**Validation:** Müge Özkan.

**Visualization:** Müge Özkan, Gerhard Wingender.

**Writing – original draft:** Müge Özkan, Gerhard Wingender.

**Writing – review & editing:** Müge Özkan, Yusuf Cem Eskiocak, Gerhard Wingender.

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
