## [Decision Letter · Decision Letter 0]

26 Mar 2021

PONE-D-21-05124

The IL-10GFP (VeRT-X) mouse strain is not suitable for the detection of IL-10 production by granulocytes during lung inflammation

PLOS ONE

Dear Dr. Wingender,

Thank you for submitting your manuscript to PLOS ONE. After careful consideration, we feel that it has merit but does not fully meet PLOS ONE’s publication criteria as it currently stands. Therefore, we invite you to submit a revised version of the manuscript that addresses the points raised during the review process.

Please address the comments of both reviewers, as outlined below, including addition of the additional FACS plots and data as requested.

We look forward to receiving your revised manuscript.

Kind regards,

Lynette Beattie, PhD

Academic Editor

PLOS ONE

Journal Requirements:

3. Please upload a new copy of Supporting Information Figure 2 as the detail is not clear.

Please follow the link for more information: https://blogs.plos.org/plos/2019/06/looking-good-tips-for-creating-your-plos-figures-graphics/

Reviewers' comments:

Reviewer's Responses to Questions

**Comments to the Author**

1. Is the manuscript technically sound, and do the data support the conclusions?

Reviewer #1: Yes

Reviewer #2: Yes

2. Has the statistical analysis been performed appropriately and rigorously? 

Reviewer #1: Yes

Reviewer #2: Yes

3. Have the authors made all data underlying the findings in their manuscript fully available?

Reviewer #1: Yes

Reviewer #2: Yes

4. Is the manuscript presented in an intelligible fashion and written in standard English?

Reviewer #1: Yes

Reviewer #2: Yes

5. Review Comments to the Author

Reviewer #1: Ozkan et al describe a potential caveat of utilising IL-10 reporter mice (VeRT-X) to identify myeloid-derived IL-10 production during an experimental model of LPS-induced lung inflammation.

The manuscript is well written, and the data is presented in a concise manner. The following comments are suggestions which I believe may add clarity to the overall message of the manuscript.

Major comments:

1. Can the authors comment on how the VeRT-X IL-10 reporter line and their results compare to any of the other eight IL-10 reporter lines, described by Bouabe 2012 (https://doi.org/10.1111/j.1365-3083.2012.02695.x)?

2. The original paper describing the VeRT-X mice (Madan et al., 2009) shows IL-10 GFP expression by B cells. Have the authors measured B cell derived IL-10 GFP levels in their model of LPS-induced lung inflammation?

3. It would be helpful to provide representative FACS plots as part of Supplementary Figure 1, to complement the gating strategy.

4. Figure 2 nicely illustrates IL-10 detection by ICCS vs GFP. It would be helpful to show representative FACS plots as part of this figure or as a Supplementary figure.

5. The figure legend for Figure 2 states: “The graph shows combined data from two independent experiments (PBS: n = 6 mice/group. LPS: n=7-9 mice/group)”. Can the authors confirm whether 6-9 mice/group were used in each individual experiment? Or whether the 6-9mice/group refers to the pooled data shown in Figure 2?

Minor comments:

Line 31: please remove ‘s’ from inflammations

Line 49: please change where to were

Line 54: please insert reference 7 after myeloid cells.

Line 54: please check reference 8, since the autofluorescence of myeloid cells was not mentioned in this reference.

Line 90-91: please clarify why data is not shown from WT controls. It might be informative to show this data as a supplementary figure.

Line 103: please remove ‘s’ from inflammations

Line 108: please remove ‘also’

Line 110: please change ‘analysis’ to analysing

Line 114: please remove ‘s’ from inflammations

Line 145: please add ‘d’ to purchase

Line 318: please superscript IL10GFP

Line 321 – 322: please superscript CD45+ CD45R/B220- CD3- Siglec-F+ F4/80+ CD11b- and IL10GFP

Reviewer #2: In this paper, the authors have proposed that IL-10 GFP reporter strain is not an appropriate model to detect IL-10 production by granulocytes. Using C57BL/6 and IL-10 GFP mice and by employing flowcytometry, the authors show that GFP signal seen in granulocytes post LPS challenge is not the actual GFP signal due to IL-10 production. Rather, it is attributable to autofluorescence. I think the authors have used proper controls in the study and results from the study illustrates important technical caveat in using GFP reporter mice for analysing myeloid cells. Overall, the study is good. However, I have few minor suggestions:

1. Line 31: Please change " during inflammations" to during inflammation. Please make this change elsewhere as well.

2. Line 39-40: Please rephrase this sentence as the meaning is not clear.

3. Line 49: Please change "lines where" to lines were.

4. Line 54: Please give reference when you discuss that IL-10 GFP strain was reported to enable identification of IL-10+

myeloid cells.

5. Line 110: Please change"analysis" to analysing.

6. Line 170: Change "where" to were.

7. Line 277-279 and 295-296: Please rephrase these two sentences as they seem incomplete.

8. Line 337-339: I think this sentence " C) Representative histograms from the spleen of PBS-------" is not required as it is a repetition of Supplementary Figure 3 legend.

9. Please check the overall grammar and sentence structure throughout the text.

6. PLOS authors have the option to publish the peer review history of their article (what does this mean?). If published, this will include your full peer review and any attached files.

Reviewer #1: No

Reviewer #2: No

---

## [Author Response · Author response to Decision Letter 0]

1 Apr 2021

Please see the point-to-point reply for details.

---

## [Decision Letter · Decision Letter 1]

27 Apr 2021

The IL-10GFP (VeRT-X) mouse strain is not suitable for the detection of IL-10 production by granulocytes during lung inflammation

PONE-D-21-05124R1

Dear Dr. Wingender,

We’re pleased to inform you that your manuscript has been judged scientifically suitable for publication and will be formally accepted for publication once it meets all outstanding technical requirements.

Kind regards,

Lynette Beattie, PhD

Academic Editor

PLOS ONE

Additional Editor Comments (optional):

Reviewers' comments:

Reviewer's Responses to Questions

**Comments to the Author**

1. If the authors have adequately addressed your comments raised in a previous round of review and you feel that this manuscript is now acceptable for publication, you may indicate that here to bypass the “Comments to the Author” section, enter your conflict of interest statement in the “Confidential to Editor” section, and submit your "Accept" recommendation.

Reviewer #1: All comments have been addressed

Reviewer #2: All comments have been addressed

2. Is the manuscript technically sound, and do the data support the conclusions?

Reviewer #1: Yes

Reviewer #2: Yes

3. Has the statistical analysis been performed appropriately and rigorously? 

Reviewer #1: Yes

Reviewer #2: Yes

4. Have the authors made all data underlying the findings in their manuscript fully available?

Reviewer #1: Yes

Reviewer #2: Yes

5. Is the manuscript presented in an intelligible fashion and written in standard English?

Reviewer #1: Yes

Reviewer #2: Yes

6. Review Comments to the Author

Reviewer #1: (No Response)

Reviewer #2: (No Response)

7. PLOS authors have the option to publish the peer review history of their article (what does this mean?). If published, this will include your full peer review and any attached files.

Reviewer #1: No

Reviewer #2: No

---

## [Editor Report · Acceptance letter]

3 May 2021

PONE-D-21-05124R1 

The IL-10^GFP^ (VeRT-X) mouse strain is not suitable for the detection of IL-10 production by granulocytes during lung inflammation 

Dear Dr. Wingender:

I'm pleased to inform you that your manuscript has been deemed suitable for publication in PLOS ONE. Congratulations! Your manuscript is now with our production department. 

Kind regards, 

on behalf of

Dr. Lynette Beattie 

Academic Editor

PLOS ONE